# Experimental Study and GRNN Modeling of Shrinkage Characteristics for Wax Patterns of Gas Turbine Blades Considering the Influence of Complex Structures

Changhui Liu [1,2], Chenghong Jiang [1], Zhenfeng Zhou [3,*], Fei Li [2,*], Donghong Wang [2] and Sansan Shuai [4]

1    School of Mechanical Engineering, Tongji University, Shanghai 200092, China;
     liuchanghui@tongji.edu.cn (C.L.); 2230216@tongji.edu.cn (C.J.)
2    School of Materials Science and Engineering, Shanghai Jiao Tong University, Shanghai 200240, China;
     wangdh2009@sjtu.edu.cn
3    College of Information Science and Engineering, Jiaxing University, Jiaxing 314001, China
4    School of Materials Science and Engineering, Shanghai University, Shanghai 200444, China
*    Correspondence: zzf@zjxu.edu.cn (Z.Z.); lifei74@sjtu.edu.cn (F.L.)

**Abstract:** With the continuous increase in power demand in aerospace, shipping, electricity, and other industries, a series of manufacturing requirements such as high precision, complex structure, and thin wall have been put forward for gas turbines. Gas turbine blades are the key parts of the gas turbine. Their manufacturing accuracy directly affects the fuel economy of the gas turbine. Thus, how to improve the manufacturing accuracy of gas turbine blades has always been a hot research topic. In this study, we perform a quantitative study on the correlation between process parameters and the overall wax pattern shrinkage of gas turbine blades in the wax injection process. A prediction model based on a generalized regression neural network (GRNN) is developed with the newly defined cross-sectional features consisting of area, area ratio, and some discrete point deviations. In the qualitative analysis of the cross-sectional features, it is concluded that the highest accuracy of the wax pattern is obtained for the fourth group of experiments, which corresponds to a holding pressure of 18 bars, a holding time of 180 s, and an injection temperature of 62 °C. The prediction model is trained and tested based on small experimental data, resulting in an average RE of 1.5% for the area, an average RE of 0.58% for the area ratio, and a maximum MSE of less than 0.06 $mm^2$ for discrete point deviations. Experiments show that the GRNN prediction model constructed in this study is relatively accurate, which means that the shrinkage of the remaining major investment casting procedures can also be modeled and controlled separately to obtain turbine blades with higher accuracy.

**Keywords:** gas turbine blade; process parameter; GRNN; shrinkage; cross-section

## 1. Introduction

With the development of aerospace, shipping, electricity, and other industries, the production scale for gas turbines is growing, and performance demands are on the increase. To better adapt to the harsh working environment, a gas turbine blade, as the critical component of the gas turbine, has been put forward with high precision, complex structure, thin wall, and a series of other requirements.

Dimensional variation control of turbine blades has always been a difficult task for the gas turbine factory. At present, the mass production of turbine blades is mainly manufactured by investment casting (IC). It is a fabrication method of the precise metal product without further processing by pouring the liquid metal into a pre-shaped pattern [1]. Howbeit, the qualification rate of the single-crystal turbine blades independently manufactured in China is less than 40%, among which the unqualified rate of the blades due to size deviation accounts for 50% [2]. It is the sophisticated IC procedures, numerous process parameters, and non-uniform shrinkage of the wax pattern and alloy that make the dimensional deviation of the turbine blade extremely difficult to control.

In the IC process, wax pattern production, ceramic shell fabrication, and casting are the three main procedures. As the first of these main procedures, wax pattern production is directly related to the dimensional accuracy of the final casting. According to the literature [3], the greatest dimensional deviation of the final casting comes from the dimensional deviation of the wax pattern. The precision of the wax pattern is impacted by the wax material, pattern geometry, and process parameters. During the research on the effects of these influence factors on the precision of the wax pattern, more attention has been paid to the influence of process parameters, such as holding pressure, holding time, and injection temperature.

Many researchers usually adopt the Taguchi approach or design of experiments (DOE) method to investigate the effects of process parameters on dimensional accuracy [4–6]. The response indicators are usually defined as dimensional shrinkage, discrete point deviations, or other performance indexes. When multiple indexes are taken as responses, the research will be combined with a multi-objective optimization algorithm [7] or converted into a single-response optimization problem with fuzzy logic [8]. Different researchers have come to different conclusions, and some of them even conflict with one another, so they have not yet been widely adopted. The relationship between the shrinkage of the wax pattern and the process parameters, it appears, has not yet been thoroughly and extensively researched [9].

When studying the relationship between process parameters and wax pattern shrinkage, researchers have omitted to consider or completely overlooked the influence of wax pattern geometry, which is one of the causes of the inconsistent results. Rezavand and Behravesh [10] extracted two design models from the blade geometry covered with the airfoil lunate (curvature) and the non-uniformity of the airfoil, respectively, and subsequently investigated the interrelationship between these two models and the process parameters and dimensional stability. The results indicated that the effects of blade curvature and non-uniform thickness are different. Hence, when studying the relationship between the shrinkage of the wax pattern and the process parameters, we have to take into account the influence of the complex structure on the shrinkage of the wax part.

With the development of deep learning methods, neural networks have been introduced to several fields related to the IC process, such as pressure casting process optimization [11–13], gating system design [14], the mechanical properties of the final casting [15], etc. Compared with using finite element or regression fitting methods to predict the dimensional variation of complex parts in the IC process, higher prediction precision can be achieved by applying neural networks appropriately.

Since it is particularly difficult to predict shrinkage in the IC process based on geometric parameters, the introduction of the neural network makes it one of the emerging research fields in the IC process. In the study of structural correlation shrinkage based on neural networks, Tian et al. [16] proposed a shrinkage prediction method for IC through the known geometric parameters of the CAD model and predicted the discrete point shrinkage for different contours. Based on a back-propagation neural network (BPNN) with input parameters of the geometric parameters of the designed I-beam castings, the final prediction accuracy they obtained is higher than that of the regression fitting method. Feng et al. [17] developed a BPNN-based prediction model between geometric parameters and radial shrinkage of the final casting based on the designed barrel casting (a hollow thin-walled casting) and also verified a higher prediction accuracy of the model compared to the regression model. To investigate the relationship between complex structural parameters and the shrinkage of hollow turbine blades, Dong et al. [18] simplified the turbine blade into a hollow thin-walled structure with resistance and non-resistance. Then, two mapping models based on a BPNN reflecting the relationship between structural parameters and the section shrinkage ratio of final casting were modeled for the transition section and common section, respectively. The average deviations were 5.8% and 2.4%, respectively, which improved the accuracy compared to existing studies. After that, the authors [19] further predicted the shrinkage of complex castings during IC based on convolutional

neural networks (CNNs) and obtained a higher precision. In contrast to the previous unpredictability, the introduction of neural networks has made it possible to clarify the coupling relationship between geometry and the shrinkage of turbine blades.

However, we note that most of the current related studies focus on the shrinkage of the final casting caused by geometric parameters. The shrinkage deformation of key procedures in the IC process, such as the wax pattern production, has only been well-studied quantitatively by a few researchers. Pattnaik et al. [20] set injection temperature, injection pressure, and holding time as input parameters. Then, they predicted the quality of the wax pattern, including linear shrinkage and surface roughness, based on a fuzzy-based artificial neural network. The predicted results are well-consistent with the experimental results, as the magnitude of the error is 3.16%. Song et al. [21] focus on reducing warpage and volume shrinkage in thin-walled parts during injection molding. By building a BPNN model optimized with a genetic algorithm and a support vector machine (SVM), they can effectively predict warpage and volume shrinkage, with predicted values of 0.93% and 1.9%, respectively, which can still be greatly improved.

Considering the expensive cost of turbine blade data acquisition, a small sample size becomes a crucial concern when introducing neural networks. From the previous studies, it can be seen that the BPNN is the most widely used in this field due to its excellent predictive performance. However, due to its nonideal performance in solving problems with small sample sizes and much noise, as well as some drawbacks such as falling into local minima and slow convergence speed [22], an alternative mathematically based statistical network, called a generalized regression neural network (GRNN), was introduced by Specht [23] as early as 1991. Its approximation ability, classification ability, and learning speed are better than the BPNN [24]. The GRNN can approximate any nonlinear function and overcome local minima with a quick-learning convergence speed due to its powerful nonlinear mapping capability and adaptable network structure. In addition, it also performs very well in prediction in cases where there is a shortage or instability of data. Thus, the GRNN has been widely used in non-linear regression problems such as noninvasive load monitoring [25,26], evaporation prediction [27], fault diagnosis [28], etc. In this paper, we attempt to build a model to predict the shrinkage of wax patterns for the complex structure of gas turbine blades using a GRNN.

In addition, to accurately model the shrinkage prediction, appropriate response indicators are essential. In previous studies, researchers have proposed various response indicators for hollow cavities, cylinders, or other specific structures, respectively. The responses include shrinkages (linear, volumetric, and contour) [16,29,30], surface roughness [20], and constrained/unconstrained dimensions [4,5]. However, the response index of shrinkage mentioned is hardly applicable to the effect of process parameters on the dimensions of the turbine blade with a variable cross-section and multi-dimensional features. This is attributed to the fact that the structure of the turbine blade, which has a variable cross-section and multidimensional and unconstrained/constrained features, results in a wide variation of dimension shrinkage in different cross-sections during wax pattern production. Thus, it is insufficient to characterize the overall dimensional change in turbine blades to adopt a specific cross-section or average dimensional shrinkage as a response. When studying the shrinkage of turbine blades, researchers have typically identified blade curvature, non-linear thickness [10], 2D discrete point deviation, and shrinkage [18,19] as response metrics. In this paper, while referring to the above classical response metrics, new response metrics are proposed to better characterize the shrinkage of turbine blades.

In summary, considering the influence of complex structures, it is the focus of this paper to quantitatively investigate the varying shrinkage characteristics of the wax patterns. To better present the wax pattern dimension change in gas turbine blades with different process parameters, we employ a new-response GRNN for training and testing. Based on the discussion above, the contribution of this paper primarily includes (1) quantifying the correlation between the process parameters and the overall shrinkage of the wax pattern via neural networks and (2) using new cross-sectional features (a combination of area features

and some discrete point deviations) as the input parameters of the GRNN to further analyze the overall shrinkage of the turbine blade.

The remainder of the paper is organized as follows: Section 2 describes an integrated method to predict the dimension changes in wax patterns. The three main modules, including DOE, the extraction method of cross-sectional features, and the introduction of a GRNN, are further elaborated in this section. In Section 3, we illustrate the experimental process and perform the data processing and the GRNN prediction modeling. After that, the result discussions, including the discrete point deviations discussion, the area and area ratio discussion, and the predicted performance analysis, will be presented in Section 4. Finally, some conclusions are discussed in Section 5.

## 2. Prediction Methodology

This section shows the details of the proposed method. Firstly, the overview of the integrated method is provided in Section 2.1, and the applied methods are described separately in detail later in the chapter. After the introduction of DOE in Section 2.2, the extraction methods of cross-sectional features are developed in Section 2.3. Finally, the introduction of the GRNN is described in Section 2.4.

### 2.1. Overview of the Integrated Method

In this paper, an integrated method to predict the wax pattern dimension change was proposed (shown in Figure 1). First, some experiments were designed and conducted, and a 3D laser scanning system named ATOS Core was used to obtain 3D point cloud data of turbine blades produced by different process parameters. Then, data preprocessing based on Geomagic Control was used to capture the representative 2D cross-sections of the turbine blade and their corresponding discrete point deviations. After the extraction of cross-sectional features was carried out, a feature prediction for given process parameters based on the trained GRNN was implemented. In future work, the outputs of the feature will be compared with the contour tolerance limits to evaluate their performance. Then, an optimization and rectification of process parameters will be adopted according to the tendency of feature outputs. It is worth noting that the focus of this paper is on GRNN modeling, training, and testing.

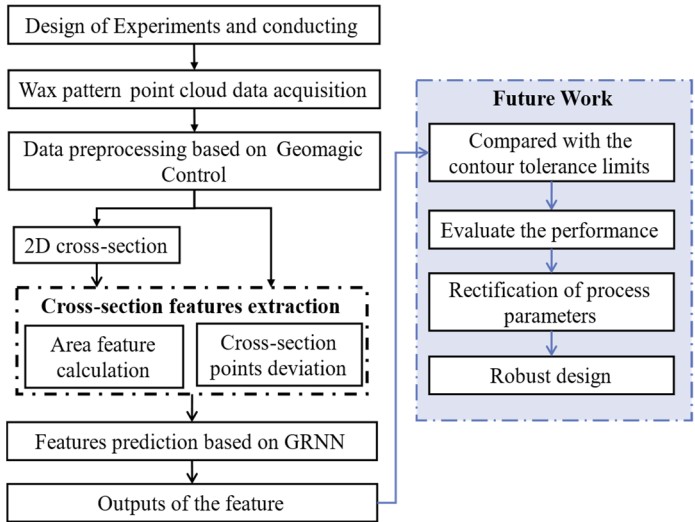

**Figure 1.** Flow chart of the proposed integrated method based on the GRNN.

### 2.2. Design of Experiments

In this paper, we introduced DOE to analyze the coupling relationship between multiple process parameters and wax pattern shrinkage. DOE is a statistical method that consists of a series of experiments conducted to understand how one or more input variables

can impact one or more output variables [31]. It helps to study the relationship between the various factors and responses. This method provides insight into the interactions that could affect the output response [32]. Based on the data generated by DOE, it is possible to develop a better predictive model.

The selection of the key process parameters is especially crucial when conducting the experimental design since there are many process parameters involved in wax pattern production, such as the mold accuracy, wax material, and wax pressing process. Pattnaik et al. [8] studied the significant effect of injection temperature, injection pressure, and injection time on the shrinkage deviation using utility concept and Taguchi method. Rezavand and Behravesh [10] used injection temperature and holding time as variable process parameters, which revealed the effect of holding time was greater than that of injection temperature. Wang et al. [33] also investigated the effect of injection parameters on the dimensional stability of wax patterns and confirmed the significant effect of holding pressure and holding time. Consequently, the injection temperature, holding pressure, and holding time were selected as input variables in this paper.

Based on the above input variables, it was required to select the specific DOE method. Depending on the different focus of the study, various DOE approaches have been derived, such as full factorial design, orthogonal array, Latin hypercube design, etc. Among them, orthogonal array (OA) is usually used as orthogonal main-effect plans in factorial experiments and statistics [34]. Thus, the orthogonal array method was selected for the design in this study.

### 2.3. Extraction Methods of Cross-Sectional Features

Because of the complex structure and large dimensions of gas turbine blades, a non-uniform shrinkage distribution is noticeable in the corresponding wax part production process. To better present the overall shrinkage of the turbine blade, the basic structure was divided into 5 cross-sections from A1 to A5, as shown in Figure 2a, and the interference of the venting holes was intentionally excluded in the selection of the cross-section. In this study, the influence of the process parameters and the complex structure on the overall shrinkage distribution are both taken into account. In addition, aiming to better evaluate the actual shrinkage change in the blade cross-sections, this study defines new cross-sectional features that contain the following three components:

- Area: As one of the measures of a geometric object, the area can effectively represent the specific shape and the deformation trend of the object. Taking area as one of the cross-sectional features not only can effectively display the state of the cross-section but also easily compare the different cross-sectional trends.
- Area Ratio: With a known CAD model of the wax pattern, each cross-sectional Target Area can be obtained; thereby, we define the Area Ratio = Area/Target Area. According to the definition, Area Ratio = 1 indicates a perfect size. Based on the value of the Area Ratio, we can determine the distance between the actual cross-section of the blade and the target cross-section. It is worth mentioning that the area features mentioned in later sections contain two features, area and area ratio.
- Discrete Point Deviation: By uniformly selecting portions of discrete points on a cross-section, the corresponding deviation value is obtained as one of the features of that cross-section. The reason for adding the discrete point deviation as one of the features is to take into account the relative offset between the cross-sections generated during the production process. It is also beneficial to keep the counterbalancing of positive and negative deviations in mind when calculating the area.

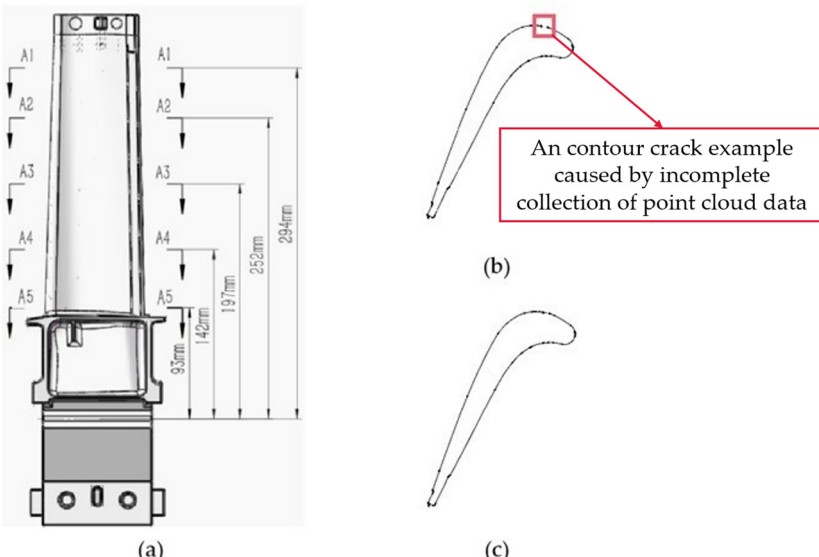

**Figure 2.** The illustration of cross-sections of the gas turbine blade: (**a**) the position of 5 cross-sections; (**b**) the original cross-section A1 of the gas turbine blade; (**c**) the cross-section A1 of the gas turbine blade after connecting cracks.

According to the type of cross-sectional features, the extraction methods are mainly divided into two categories: the area features calculation and extraction of point deviation. The 2D cross-sections and their corresponding discrete point deviations can be easily obtained using the Geomagic Control software, while the irregular shape of the cross-sections makes the extraction of the area features a key problem for feature extraction. Various methods are currently used to measure image area, such as the square grid method [35], photoshop-based image area measurement, pixel-based image processing methods [36–38], etc. Among them, the pixel-based method is quite simple, efficient, and highly accurate, so the pixel-based method was elected to calculate the blade cross-sectional area in this study.

Figure 3 shows the extraction process for area features. The details are elaborated as follows:

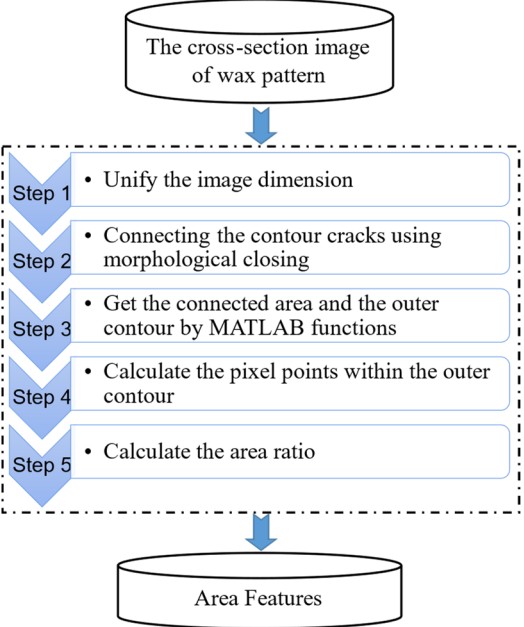

**Figure 3.** The flow chart of area features extraction.

Step 1: Unify the dimensions of all cross-section images. The area values represented by the number of pixels in different-sized images will be significantly different, while the size of the images exported in the Geomagic Control software is inconsistent.

Step 2: By applying the morphological closing operator [39], the unclosed borders of parts are retrieved and connected. Since the cross-sections of the experimental group are obtained from the 3D point cloud data, the obtained cross-sections may have some small cracks and do not form a closed geometry. The cross-section A1 graphs of the wax pattern for the gas turbine blade before and after connecting cracks are shown in Figure 2b,c, respectively.

Step 3: Obtain the connected area by using the BWLABEL function in MATLAB and obtain the outer contour of the image by using the BWSELECT function.

Step 4: Calculate the pixel points within the contour, i.e., the area of the section.

Step 5: After the cross-sectional areas of both the wax pattern of the experimental group and the corresponding CAD model are solved, the area ratio of each experimental group is calculated separately.

### 2.4. Generalized Regression Neural Network (GRNN)

The topological structure of the gas turbine blade shrinkage rate prediction model based on the GRNN is shown in Figure 4. The input parameters include some process parameters and other needed input parameters. The cross-sectional features of the turbine blade are used as the output. In this predictive model, we define the input-layer matrix of the network as $X = [x_1, x_2, \cdots, x_n]$ and the output-layer matrix of the network as $Y = [y_1, y_2, \cdots, y_k]$. The number of neurons in the pattern layer is equal to the dimension of input variables, set as $n$. The number of neurons in the summation layer is equal to the dimension of output variables plus one, set to $k + 1$. The transfer function of the pattern layer is expressed in Equation (1). For the summation layer, there are two summation functions [40], namely, $S_D$ and $S_{Nj}$, which are expressed in Equations (2) and (3). The difference is that the former performs an arithmetic summation of the output of all pattern-layer neurons directly, while the latter is a weighted sum of all neurons in the pattern layer with the connection weight of the $j$th element in the $i$th output sample. Dividing the two output types of the above summation layer, we can obtain the model output in Equation (4).

$$P_i = exp\left[\frac{(X - X_i)^T(X - X_i)}{2\sigma^2}\right], \ i = 1, 2, \cdots, n \tag{1}$$

$$S_D = \sum_{i=1}^{n} P_i \tag{2}$$

$$S_{Nj} = \sum_{i=1}^{n} y_{ij} P_i, \ j = 1, 2, \cdots, k \tag{3}$$

$$y_j = \frac{S_{Nj}}{S_D}, \ j = 1, 2, \cdots, k \tag{4}$$

where $X$ is the input, $X_i$ is the learning sample corresponding to the $i$th neuron, and $\sigma$ is the radial basis expansion coefficient: SPREAD.

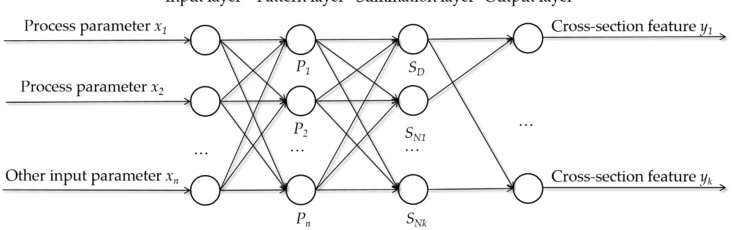

**Figure 4.** The topological structure of the GRNN.

## 3. Application of Methodology

To validate the proposed method, some experiments were conducted (see Section 3.1). In this section, the experimental design is first carried out so that more data based on a gas turbine blade can be created. After the data acquisition mentioned in Section 3.2 is conducted, the extraction of area features is implemented. Finally, the GRNN modeling phase is performed in Section 3.3.

### 3.1. Orthogonal Array Design Phase

Based on the critical process parameters selected in Section 2.2, including holding pressure, holding time, and injection temperature, we give the designated symbols and ranges in Table 1. Referring to the parameter levels in Table 1, the design of experiments based on three control factors is constructed (orthogonal arrays) in Table 2.

**Table 1.** Process parameters and their values at different levels.

| Symbol | Process Parameters | Unit | Range | Level 1 | Level 2 | Level 3 |
|--------|--------------------|------|-------|---------|---------|---------|
| A | Holding pressure | bar | 12–18 | 12 | 15 | 18 |
| B | Holding time | s | 150–210 | 150 | 180 | 210 |
| C | Injection temperature | °C | 56–62 | 56 | 62 | - |

**Table 2.** Experimental layout.

| Trial No. | A (bar) | B (s) | C (°C) |
|-----------|---------|-------|--------|
| 1 | 2 | 2 | 1 |
| 2 | 2 | 3 | 2 |
| 3 | 1 | 3 | 1 |
| 4 | 3 | 2 | 2 |
| 5 | 3 | 1 | 1 |
| 6 | 1 | 2 | 2 |
| 7 | 2 | 1 | 2 |
| 8 | 1 | 1 | 1 |
| 9 | 3 | 3 | 1 |

According to the experimental matrix design in Table 2, we conducted 9 groups of wax pattern experiments with different process parameter levels. Based on the wax pattern production steps [41], the produced wax patterns were arranged in order as shown in Figure 5.

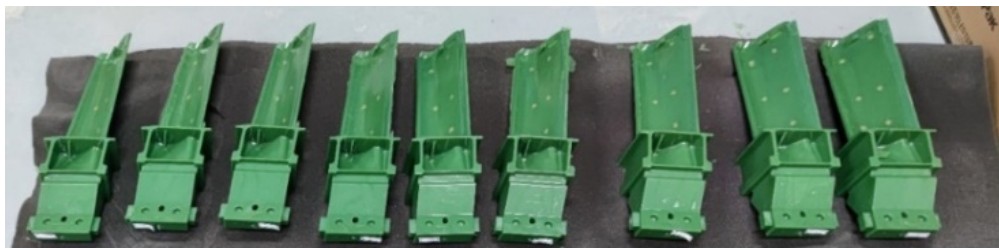

**Figure 5.** Wax patterns of a gas turbine blade.

### 3.2. Data Processing

In this section, the data acquisition mode of wax patterns for a practice gas turbine blade is first introduced, and relevant data is obtained in Section 3.2.1. After that, the extraction of area features is performed in Section 3.2.2.

#### 3.2.1. Data Acquisition

Following the production of 9 groups of experimental wax patterns, the ATOS Core, a 3D laser scanning system, was used to obtain the integral 3D wax pattern point cloud data

by taking multiple photos and stitching multiple image data under the same coordinate system. The wax patterns of the experiments were scanned, and the 3D point cloud data of experimental wax patterns (from No. 1 to No. 9, where No. X represents the wax pattern of Trial No. X in Table 2) were compared with the CAD model, as shown in Figure 6. The overall deviation of the wax pattern varies according to the process parameters. Different colors refer to different deviation values. From the image, it is clearer that while the deviation of the leaf body is lower, the top and tenon regions of the wax pattern produce larger deviations. The former has a prominent color of green and an absolute deviation within 0.1 mm, while the latter two have dominating colors of red and blue, respectively, with an absolute deviation of about 0.5 mm. However, because the body portion is created in a single operation, while the top and tenon portions still need to be machined afterward, its dimensional variation needs to be tightly controlled to within plus or minus 0.5 mm. It can be noticed that the 1st, 2nd, and 4th groups have fewer deviations when comparing the 3D point cloud deviations of the body portions of the 9 groups of wax patterns. This indicates that these three groups' process parameters are better adapted, and the resulting wax patterns are higher in accuracy.

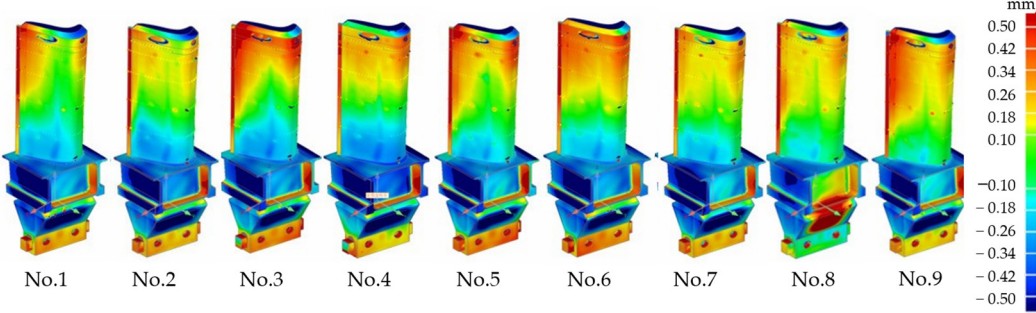

**Figure 6.** 3D point cloud data of experimental wax patterns (from No. 1 to No. 9, where No. X represents the wax pattern of Trial No. X in Table 2) compared with the CAD model. Different colors refer to different deviation values.

In the Geomagic Control software, the point cloud data can be inspected and analyzed. By comparing the acquired point cloud data with the CAD model of the blade wax pattern, the 2D section deviation can be obtained by setting the cross-section. The division of the cross-sectional contour curve for 2D section A1 and its partial discrete point deviations of the 1st group are shown in Figure 7. It is worth mentioning that the discrete points of each cross-section in this study are selected by equal numbers and equal distances on the outer contour of the wax pattern for a turbine blade. The distribution of discrete points starts from the Trailing edge (TE), then goes through the Suction side (SS), the Leading edge (LE), the Pressure side (PS), and back to the TE in turn. Note that the discrete point deviation values are highly correlated with the region of the section contour curve and that the trend along the contour varies from region to region. Further quantitative analysis of the discrete point deviations for each 2D cross-section will be developed in Section 4.1.

### 3.2.2. Extraction of Area Features

According to the method described in Section 2.3, the 2D cross-section images of the CAD model were first unified with a pixel size of 936 × 671. Then, we connected the unclosed border parts and obtained the corresponding outer contours. Finally, the cross-sectional area values (in pixels) of the CAD model from A1 to A5 are calculated as 44,696, 49,808, 57,614, 66,849, and 76,821, in that order. Similarly, the cross-sectional area for all practice wax patterns and the corresponding area ratio is calculated and shown in Table 3.

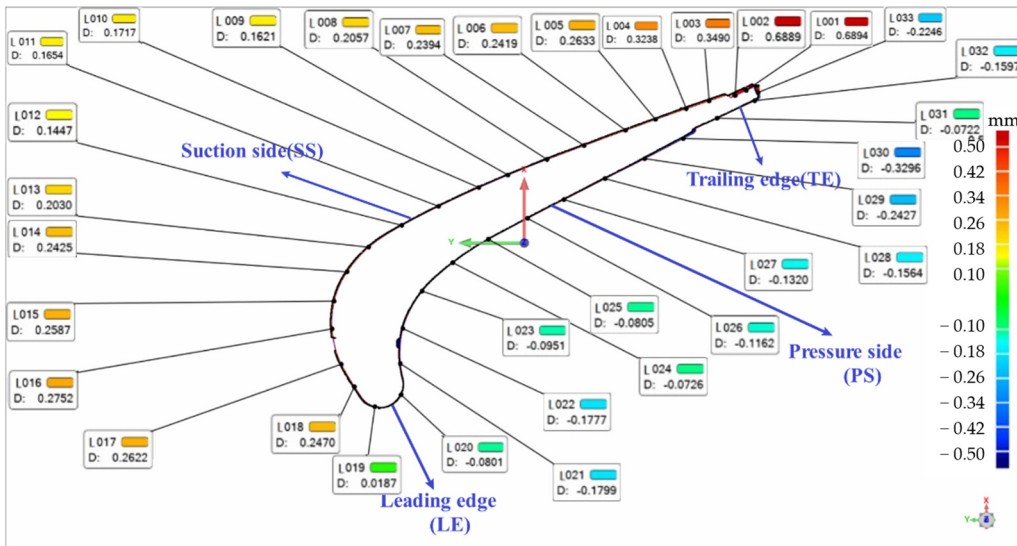

**Figure 7.** The division of the cross-sectional contour curve for 2D section A1 and its partial discrete point deviations of trial No. 1 compared with the CAD model. There are 33 points evenly distributed on the contour of the section, where "L001" refers to point 1, and its corresponding "D: 0.6894" refers to a deviation of 0.6894 mm.

**Table 3.** Calculation results of the area features for all practice wax patterns.

| No. | Area (in Pixels) | | | | | Area Ratio | | | | |
|---|---|---|---|---|---|---|---|---|---|---|
| | A1 | A2 | A3 | A4 | A5 | A1 | A2 | A3 | A4 | A5 |
| 1 | 45,439 | 50,775 | 58,962 | 68,095 | 78,311 | 1.0166 | 1.0194 | 1.0234 | 1.0186 | 1.0194 |
| 2 | 45,389 | 50,516 | 58,736 | 67,955 | 78,147 | 1.0155 | 1.0142 | 1.0195 | 1.0165 | 1.0173 |
| 3 | 46,330 | 51,444 | 59,343 | 68,437 | 78,539 | 1.0366 | 1.0328 | 1.0300 | 1.0238 | 1.0224 |
| 4 | 45,665 | 50,642 | 58,664 | 67,845 | 77,988 | 1.0217 | 1.0167 | 1.0182 | 1.0149 | 1.0152 |
| 5 | 45,925 | 51,024 | 59,336 | 68,805 | 78,603 | 1.0275 | 1.0244 | 1.0299 | 1.0293 | 1.0232 |
| 6 | 46,183 | 51,159 | 59,086 | 68,235 | 78,550 | 1.0333 | 1.0271 | 1.0255 | 1.0207 | 1.0225 |
| 7 | 45,855 | 50,921 | 59,003 | 68,219 | 78,340 | 1.0259 | 1.0223 | 1.0241 | 1.0205 | 1.0198 |
| 8 | 45,873 | 51,027 | 59,195 | 68,752 | 78,468 | 1.0263 | 1.0245 | 1.0274 | 1.0285 | 1.0214 |
| 9 | 46,277 | 51,368 | 59,337 | 68,354 | 78,294 | 1.0354 | 1.0313 | 1.0299 | 1.0225 | 1.0192 |

*3.3. GRNN Modeling Phase*

This section will develop a prediction model for the wax patterns of the gas turbine blades' shrinkage characteristics in relation to the structural features and process parameters. Among the structural features, we highlight the variable cross-sections and multi-dimensional features of the gas turbine blade. Thus, five cross-sections from A1 to A5 are divided as described in Section 2.2.

Based on the above experiments, we obtained 45 samples for the training and testing of the GRNN model. Notably, the area features obtained from different cross-sections (as shown in Table 3) are significantly different. We divided the obtained samples into 9 groups according to the process parameters when performing the GRNN model training and testing. Each group contains the cross-sectional features of five cross-sections. Since the neural network needs more data for feature learning, we took 8 groups of them randomly as training sets, and the remaining group was used as verification samples to test the performance of the prediction model.

The parameters setting of the GRNN prediction model is given as follows: The input-layer nodes are set to 4, namely, the holding pressure (A), the holding time (B), the injection temperature (C), and the section height (D). Among them, the section height (D) is the key structural parameter of the turbine blade. The number of nodes in the output layer is set to

34, which contains the area, area ratio, and 32 discrete point deviations. The kernel function of the GRNN algorithm is the Gaussian radial basis kernel function.

In addition, the value of SPREAD is quite critical because it is the only parameter of the GRNN model that improves performance. When it is too large, it increases the complexity of the computation and results in an overly smooth output. Contrarily, when it is too small, there are not sufficient neurons involved in the computation, which in turn brings about poor generalization [42]. Generally, the SPREAD was optimized by a trial-and-error method in the present study. In this paper, a combination of k-fold cross-validation (CV) and the trial-and-error method was used to optimize the SPREAD because this method can effectively avoid overfitting and underfitting and guarantee high prediction accuracy when the data volume is small.

The k-fold CV method is a resampling technique. The initial sample is randomly divided into *k* subsamples of equal size. The remaining fold is then used for evaluation after a model has been trained on $k - 1$ of the folds (subsamples). Each of the k folds is utilized as the validation data exactly once during the k times this process is conducted. In the k-fold CV process of this study, *k* was set to 5. The iteration time of the cycle was 200, and the mean absolute percent error (MAPE) was used as the performance index to select the optimal SPREAD. After training the prediction model, the testing of cross-sectional features was conducted.

## 4. Results Discussion

We discuss the results of the above experiments in this section. The discrete point deviations, area shrinkage features, and the predicted model of the gas turbine blade are discussed in the next sections.

### 4.1. Discussion for Discrete Point Deviations

In this section, the discrete point deviations are qualitatively analyzed as influenced by the process parameters and the position of the cross-section, respectively. According to the variation of each discrete point and its corresponding position, we can make a better qualitative analysis of the shrinkage of the blade in each section.

Firstly, under the five cross-sections, we sequentially plot box plots of the fluctuations of discrete point deviations for nine groups of process parameters, as shown in Figure 8. From the observation in Figure 8, we can summarize the following phenomena:

(1) The structure at different parts of the contour has shown significant variation, and there is a significant difference in the deviation of different cross-sections. For example, in Figure 8a, the deviation of the SS section is significantly higher than that of the PS section, while in Figure 8d, the deviation of the SS section is significantly lower than that of the PS section.

(2) As the height of the section decreases, that is, from the top to the tenon of the blade, only the deviation value of the SS segment gradually decreases, while the deviation value of the remaining segments increases. Notably, the transition from section A4 to A5 shows an abrupt decreasing trend in deviation for the LE and PS segments.

(3) The effects of the various process parameters at different positions on the cross-section contour varies significantly and for different cross-sections as well. For example, in Figure 8c, the impact of process parameter fluctuations on the SS section is much smaller than that on the TE and SS sections, while in Figure 8e, the deviations at all points do not have large fluctuations when process parameters change.

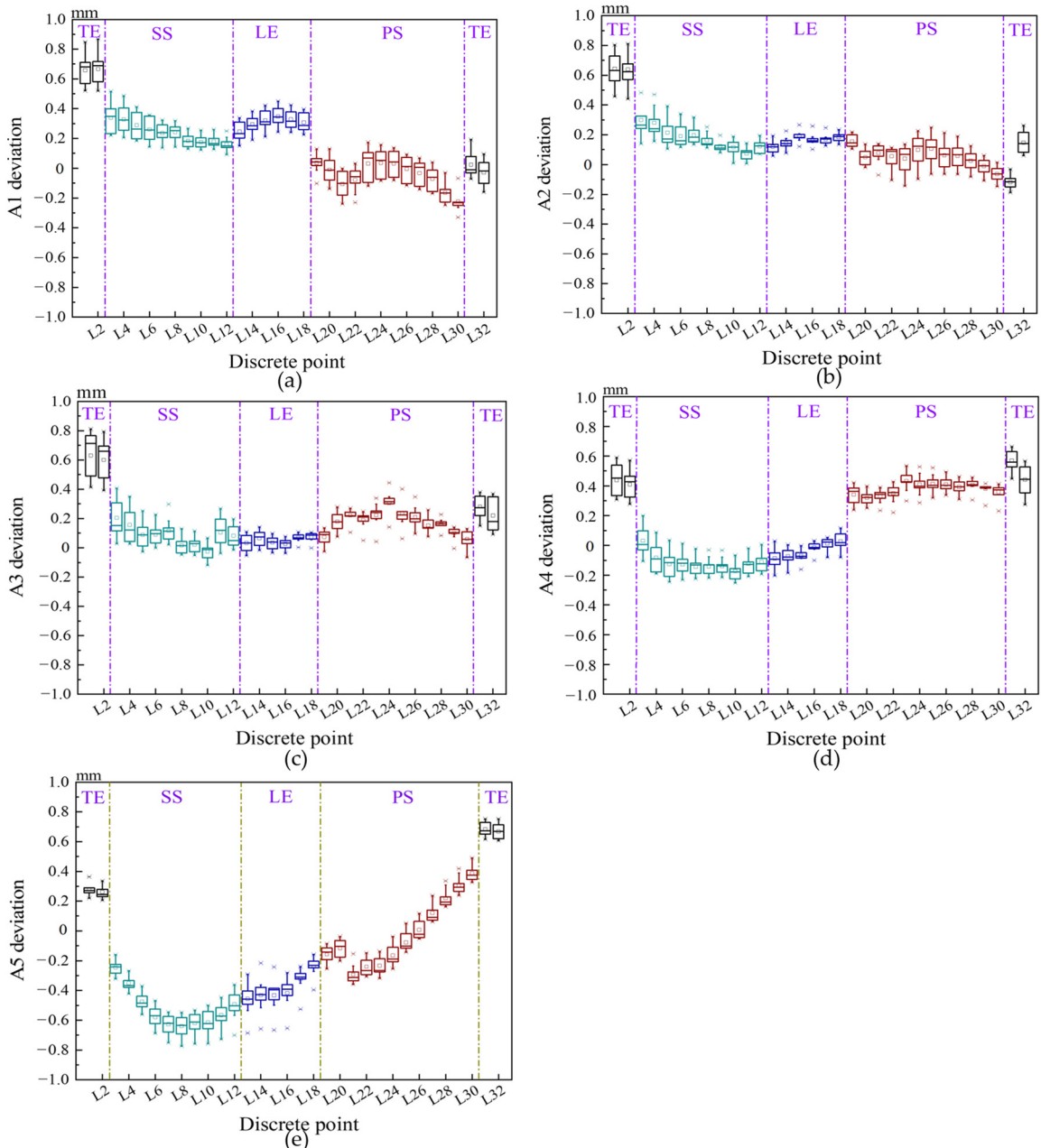

**Figure 8.** Box plots of the discrete point deviations of different sections for 9 groups of experiments: (**a**) Section A1; (**b**) Section A2; (**c**) Section A3; (**d**) Section A4; (**e**) Section A5.

### 4.2. Discussion for Area and Area Ratio

The bar graph of the area and the box plot of the area ratio corresponding to Table 3 are plotted in Figure 9. In Figure 9a, the area value of each cross-section increases continuously from A1 to A5, and the values of the experiments are all above the target value. However, we can observe that the area gap values due to different process parameters still have variability. Figure 9b shows the details of the different process parameters leading to different area ratios, among which the area features of the second and fourth groups are closest to the target values. Moreover, an obvious difference exists in the fluctuations of the area ratios produced by the different process parameters for the different cross-sections. This is similar to the results observed in Section 4.1. The dispersion degree of area ratios of the third and ninth groups is significantly larger than that of the other groups.

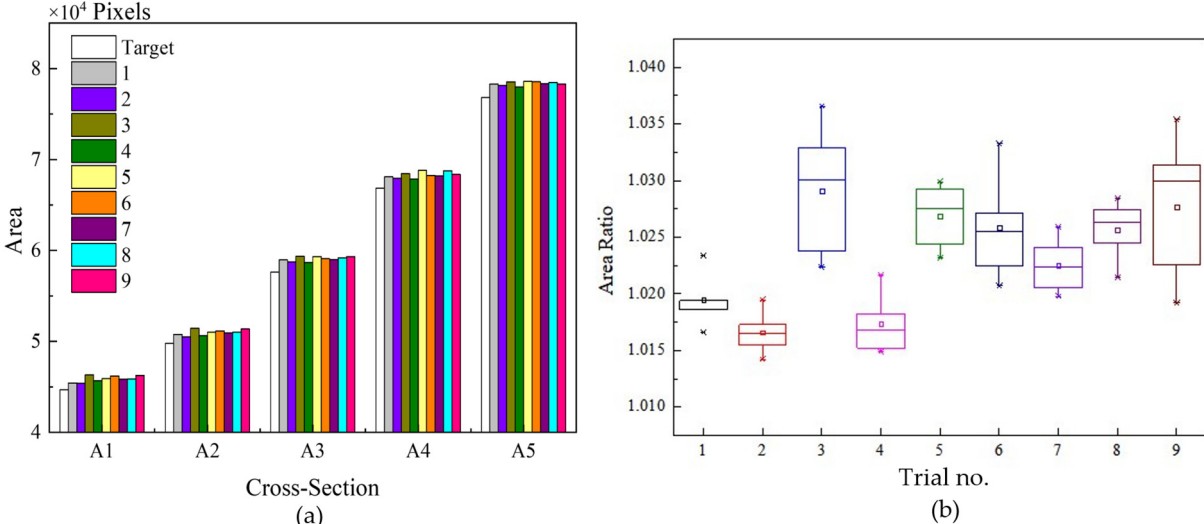

**Figure 9.** Area feature results: (**a**) bar graph of the area calculation results; (**b**) box plot of the area ratio results.

### 4.3. Predicted Model Analysis

To validate the developed GRNN prediction model, some experiments were conducted. Next, we will discuss two aspects of the prediction results and prediction performance.

#### 4.3.1. Prediction Results

When we take the data of the fourth group as the testing sample, the prediction results of area features and discrete point deviations of the A1 cross-section are shown in Figure 10. As can be seen from Figure 10, the prediction results of our proposed algorithm for the cross-sectional features of the gas turbine blade are matched with the actual values. It shows that the GRNN prediction model constructed in this study is relatively accurate. From the analysis of the predicted results in Figure 10b, it is clear that the deviation has changed abruptly due to the abrupt change in the casting structure at discrete points 3, 13, 19, and 31, which also verifies the validity of the model.

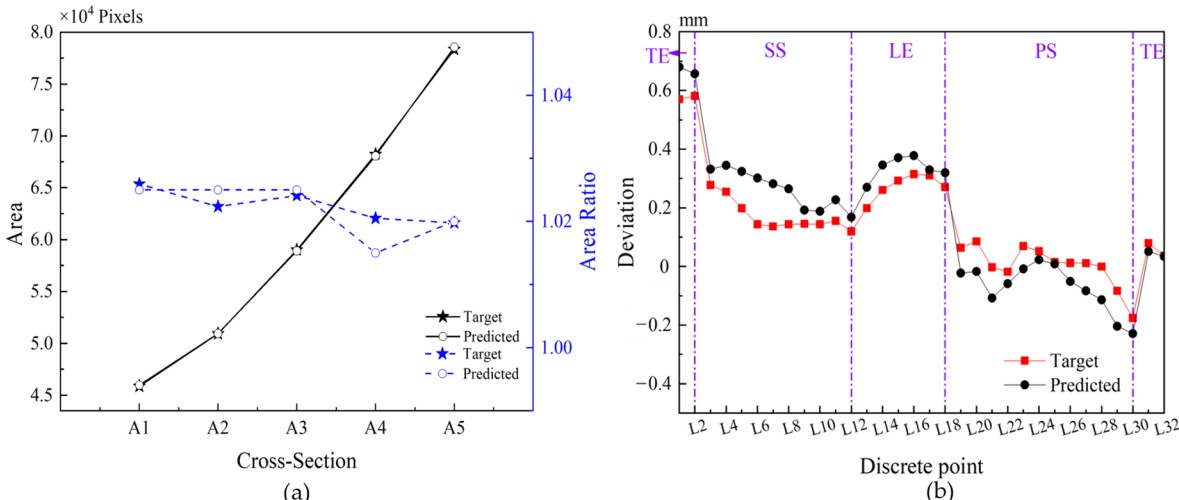

**Figure 10.** Prediction results based on the GRNN: (**a**) area features; (**b**) discrete point deviations of the A1 cross-section.

### 4.3.2. Prediction Performance

With the prediction model already trained, further experiments are needed to better determine the performance of this model. We utilized 5-fold cross-validation and trial-and-error to determine the parameter OPTIMAL SPREAD in the training, described in Section 3.3. To address the problem of the small amount of data in this study, 5-fold cross-validation was used again with optimal SPREAD. We selected one group of data as the testing sample in turn and used the remaining groups of data as the training sample. Nine trials were thus conducted, and the pros and cons of the model were measured by taking the combination of multiple performance evaluation functions.

We used relative error (RE), mean square error (MSE), and MAPE as error indicators for analysis. The formulas for the three indicators are given in equations as follows:

$$RE = \frac{|Y_i - T_i|}{T_i} \times 100\% \tag{5}$$

$$MSE = \frac{1}{k} \sum_{i=1}^{k} (Y_i - T_i)^2, \ j = 1, 2, \cdots, k \tag{6}$$

$$MAPE = \frac{1}{k} \sum_{i=1}^{k} \frac{|Y_i - T_i|}{T_i}, \ j = 1, 2, \cdots, k \tag{7}$$

where, $T_i$ is the target value corresponding to the $i$th neuron, and $Y_i$ is the testing sample corresponding to the $i$th neuron.

The RE can effectively reflect the deviation of the predicted result from the target value and determine the confidence of the predicted result. The MSE is the average of the sum of squared prediction errors, avoiding the problem that positive and negative errors cannot be summed. The MAPE is a percentage value, and it is generally considered that the prediction accuracy is great when MAPE is less than 10%.

Since the discrete point deviation may be zero, it is not reasonable to use RE or MAPE to evaluate its prediction performance. Therefore, in the following analysis, we use RE and MAPE to evaluate how well the model predicts area features. MSE is used to evaluate the model's prediction performance for discrete point deviations. The RE table and the corresponding box plot of prediction for different cross-sectional area features are shown in Table 4 and Figure 11, respectively. The average RE of the area is 1.5%, and the average RE of the area ratio is 0.58%. The REs of the areas of sections A1 and A5 are larger than that of other sections according to Figure 11, while the REs of the area ratios are more average and smaller overall, with the REs of section A5 being the smallest.

**Table 4.** RE table for different cross-sectional area features based on the GRNN model.

|  | **A1** | **A2** | **A3** | **A4** | **A5** | **Average RE** |
|---|---|---|---|---|---|---|
| Area | 2.3646% | 1.2671% | 0.4978% | 1.0554% | 2.3196% | 1.5009% |
| Area Ratio | 0.9733% | 0.7518% | 0.7053% | 0.4429% | 0.0365% | 0.5820% |

In addition, a comparison of area features with a BPNN using MAPE as the performance evaluation function is shown in Figure 12. Figure 12 shows that the prediction accuracy of the area ratio is superior to that of the area. Moreover, compared to the BPNN, the MAPE of the GRNN in predicting area is improved.

After the error analysis of area features, it is also necessary to analyze the prediction error of discrete point deviations. Based on the predicted values of each discrete point of the nine training trials, the corresponding MSEs are calculated and plotted as box line plots, as shown in Figure 13a. In addition, a comparison of discrete point deviations with the BPNN using MSE as the performance evaluation function is drawn in Figure 13b. From Figure 13, we can observe that the MSEs of the central discrete points are significantly less than the MSEs of the discrete points on the two sides. The apparent difference in prediction

error for different discrete points partially depends on the variation in the dispersion of their original data. Moreover, the maximum MSE of discrete point deviations based on the GRNN is less than 0.06 mm$^2$, while the maximum MSE based on the BPNN is nearly 0.2 mm$^2$. The experiments show that the GRNN prediction model constructed in this study is relatively accurate.

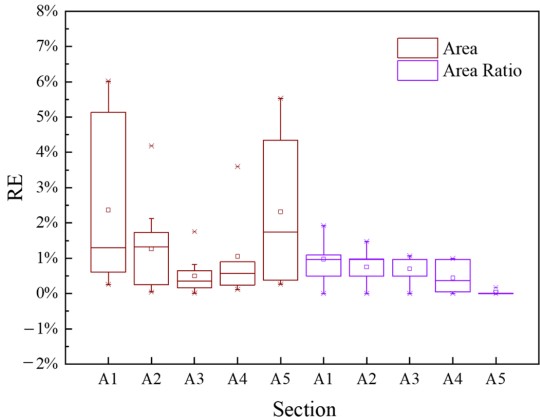

**Figure 11.** RE box plot of area features based on the GRNN model.

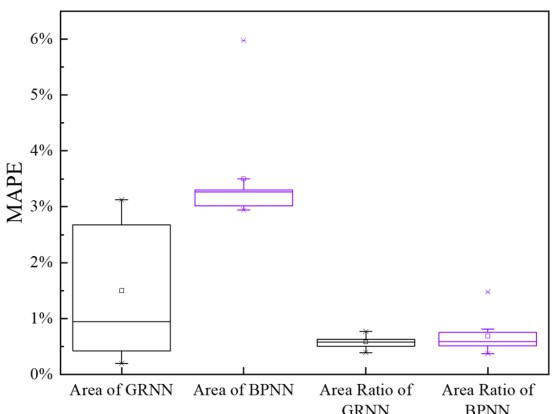

**Figure 12.** MAPE box comparison plot based on the area features of the GRNN and BPNN.

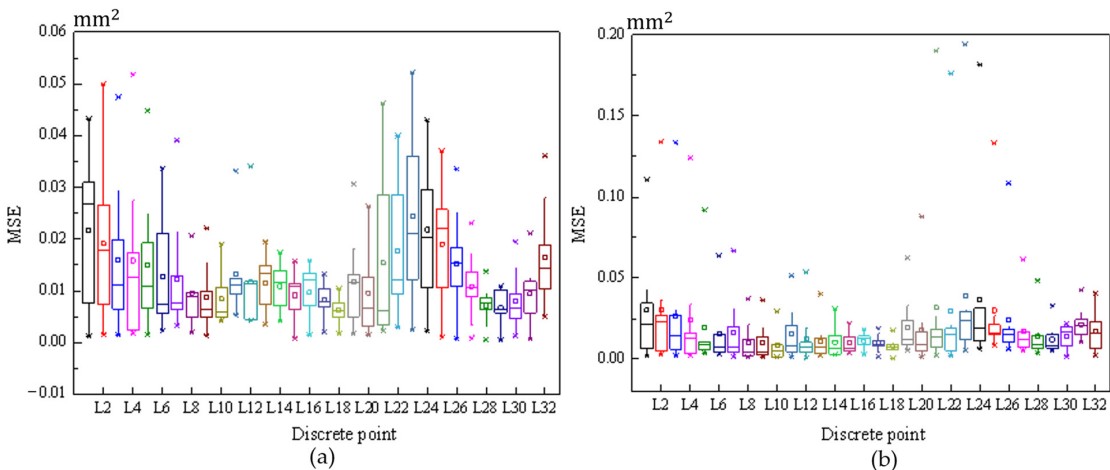

**Figure 13.** The MSE box plot of discrete point deviations: (**a**) based on the GRNN; (**b**) based on the BPNN.

## 5. Conclusions

In this study, new turbine blade cross-sectional features are defined, and a GRNN-based approach is proposed to predict the wax pattern shrinkage of the turbine blade. The related work and corresponding conclusions are as follows:

(1) Defined the area features including area and area ratio and combined the common discrete point deviation to form a comprehensive index, namely, cross-sectional features, which better reflects the shrinkage of the blade cross-section. The pixel point-based area calculation method is used to extract the area features. The results of the area features extraction were analyzed, and it is concluded that the second and fourth groups of experimental wax patterns are closer to the CAD model of the wax pattern.

(2) In Section 4.1, the coupling interactions between discrete point deviations and section positions and process parameters are qualitatively analyzed. It is finally concluded that the accuracy of the fourth group of experimental wax patterns is optimal, which corresponds to a holding pressure of 18 bar, a holding time of 180 s, and an injection temperature of 62 °C. In Section 4.2, a consistent result is obtained with that in Section 4.1. That is, the fluctuation of the area ratios resulting from the various process parameters for the various cross-sections shows a clear difference.

(3) A generalized regression neural network (GRNN) model was established to predict the wax pattern shrinkage of the gas turbine blade under different process parameters in this study. A combination of cross-validation and trial-and-error methods was used to train and predict small data (only 45 samples of data). Finally, different performance evaluation functions were applied to measure the accuracy of this prediction model, resulting in an average RE of 1.5% for the area, an average RE of 0.58% for the area ratio, and a maximum MSE of less than 0.06 $mm^2$ for the discrete point deviations. The prediction accuracy of the model will be improved in the future if more data can be acquired. Based on this accurate predictive model of wax pattern shrinkage, we can further make a robust design to optimize and rectify process parameters in the future.

(4) Without considering the influence of the pattern material on the shrinkage change, the proposed GRNN model can relatively accurately predict the wax pattern shrinkage change in turbine blades with a complex structure, which indicates that we can also model and control the shrinkage of the remaining main procedures of the investment casting separately by applying the proposed method and thus finally obtain turbine blades with higher accuracy.

**Author Contributions:** Conceptualization, C.L., Z.Z. and F.L.; formal analysis, C.L. and C.J.; funding acquisition, C.L. and Z.Z.; investigation, C.L., C.J., D.W. and S.S.; methodology, C.L. and C.J.; software, C.L. and C.J.; supervision, C.L., Z.Z. and F.L.; writing—original draft, C.L. and C.J.; writing—review and editing, C.L., Z.Z. and F.L. All authors have read and agreed to the published version of the manuscript.

**Funding:** This research is sponsored by the Natural Science Foundation of Shanghai, Grant No. 22ZR1463900, Zhejiang Provincial Science and Technology Plan Project (Lingyan), Grant No. 2022C03121, the Fundamental Research Funds for the Central Universities, Grant No. 22120220649, and the Research Project of State Key Laboratory of Mechanical System and Vibration, Grant No. MSV202318.

**Data Availability Statement:** The data that support the findings of this study are available upon reasonable request.

**Conflicts of Interest:** The authors declare no conflict of interest.

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
