# Peer review of "Experimental Study and GRNN Modeling of Shrinkage Characteristics for Wax Patterns of Gas Turbine Blades Considering the Influence of Complex Structures"

_machines, doi:10.3390/machines11060645_

Round 1

Reviewer 1 Report

The title of the paper suggests that an Experiment Study will be presented, but there are very few research results. If the authors' analysis was based on experimental research, the stand on which the research was performed should be presented and the research results should be documented accordingly. Figure 6 is for reference only. The measurement results should include an uncertainty analysis, especially since they are used to train the model.

Figure 8 is not clear how to interpret the values on the deviation axis. If 1.0 corresponds to 100% of the dimension, then the values of sequentially plot box plots are very large, so it should be interpreted accordingly. If it means something else, it should be clearly defined.

The content does not indicate whether the trained model was verified with a different set of data apart from the training set. When we train and test a neural network on the same set, the results are usually very good, but it does not always have to be so for other data for which the network has not been trained.

English is understandable to me. Like the authors, English is not my native language and I would express some sentences in other words, but this is undoubtedly subjective.

Reviewer 2 Report

The topic is really interesting. The manuscript needs some modification to be ready for publication.

- The references need modification, and more new (2018-2023) references should be added to the manuscript.

- The content, axis label and numbers in the figures (particularly figures 6, 7 & 10) need modification. They are not readable.

- Could you please clarify the source of errors in your prediction model?

- How accurate is the model, and how considering the pattern materials will affect the accuracy?

- How could it be possible to use a model like LSTM, and what could happen to the final prediction from your point of view?

- What is your principal weakness in this prediction model?

- In Figure 6, what is the wax pattern and how the colour difference shows it?

- there are some typos in the manuscript.

The English language is good enough. There are some typos in the manuscript.

Round 2

Reviewer 2 Report

The authors implemented extensive changes in the manuscript, which I believe improved the quality to reach an acceptable level for publication.  Considering all changes, the manuscript is ready for the next level to follow the publishing process.